# Energy vs. Nutritional Potential of Virginia Mallow (*Sida hermaphrodita* L.) and Cup Plant (*Silphium perfoliatum* L.)

**DOI:** 10.3390/plants11212906

**Published:** 2022-10-28

**Authors:** Jona Šurić, Jana Šic Žlabur, Anamarija Peter, Ivan Brandić, Sandra Voća, Mia Dujmović, Josip Leto, Neven Voća

**Affiliations:** 1Department of Agricultural Technology, Storage and Transport, Faculty of Agriculture, University of Zagreb, Svetošimunska Cesta 25, 10000 Zagreb, Croatia; 2Department of Field Crops, Forage and Grassland, Faculty of Agriculture, University of Zagreb, Svetošimunska Cesta 25, 10000 Zagreb, Croatia

**Keywords:** renewable energy sources, agriculture biomass, specialized metabolites, individual polyphenolics, antioxidant capacity

## Abstract

The world today faces several pressing challenges: energy from non-renewable sources is becoming increasingly expensive, while at the same time the use of agricultural land for food production is decreasing at the expense of biofuel production. Energy crops offer a potential solution to maximizing the use of land. In order to provide new value to the by-product, it is necessary to investigate its possible nutritional and functional potential. Therefore, the main objective of this study was to determine the energetic, nutritional, and functional potential of the species *Sida hermaphrodita* L. and *Silphium perfoliatum* L. in different phenophases. The analyzed energy potential of the mentioned species is not negligible due to the high determined calorific value (17.36 MJ/kg for Virginia mallow and 15.46 MJ/kg for the cup plant), high coke content (15.49% for the cup plant and 10.45% for Virginia mallow), and desirably high carbon content, almost 45%, in both species. The phenophase of the plant had a significant influence on the content of the analyzed specialized metabolites (SM) in the leaves, with a high content of ascorbic acid at the full-flowering stage in Virginia mallow (229.79 mg/100 g fw) and in cup plants at the end of flowering (122.57 mg/100 g fw). In addition, both species have high content of polyphenols: as much as 1079.59 mg GAE/100 g were determined in the leaves of Virginia mallow at the pre-flowering stage and 1115.21 mg GAE/100 g fw in the cup plants at the full-flowering stage. An HPLC analysis showed high levels of ellagic acid and naringin in both species. In addition, both species have high total chlorophyll and carotenoid concentrations. Due to their high content of SM, both species are characterized by a high antioxidant capacity. It can be concluded that, in addition to their energetic importance, these two plants are also an important source of bioactive compounds; thus, their nutritional and functional potential for further use as value-added by-products should not be neglected.

## 1. Introduction

The instability of fossil fuel market prices is leading to increased interest in renewable energy, especially in the production of biofuel from alternative sources. According to 2019 data, in the European Union, about 23% of the total cereal crops grown are used for food production, while about 12% are used for biofuel and energy production. In addition, about 55% of the calories produced from food crops worldwide are directly intended for human consumption [1,2]. It is estimated that intensive biofuel production may negatively impact food production due to insufficient agricultural land. At a time when non-renewable energies are becoming very expensive and increasingly difficult to access, it is necessary to find new natural, renewable resources [3]. In addition to the energy crisis, the world is also facing a crisis regarding the general availability of food for human consumption, as well as an increasingly pressing problem of waste disposal. Therefore, it is important to apply sustainable development practices that, among other things, provide new added value to the by-product; increasingly apply recovery practices to obtain new products; and, finally, recycle and reuse the raw material so that its usability is maximized [4,5]. In anticipation of an integrated energy and land use policy for food production, the use of the biomass produced for other purposes is one of the possible solutions to the above problem.

What are energy crops anyway? They are annual or perennial, fast-growing species characterized by the fact that they produce a large amount of biomass per unit area and are primarily intended for biogas production or direct combustion [6]. Since energy crops are also a plant species with a specific phytochemical profile, their nutritional composition should not be neglected; therefore, some other uses should be identified before their conversion to energy. Virginia mallow (*Sida hermaphrodita* L.) and the cup plant (*Silphium perfoliatum* L.) have recently received increasing research attention for their energy value as potential renewable energy sources [7]. Both plants are native to North America and were mainly explored as animal feed and energy crops in the 1980s [8]. Nowadays, Virginia mallow and cup plants can be reconsidered as potential energy sources for biofuel production or direct combustion. In addition to their energy potential, both species have high ecological potential in terms of biodiversity conservation; sequestration, i.e., the long-term storage of CO_2_; and soil-quality regulation, especially soil erosion reduction, and are referred to as phytoremediation plants [9]. It is important to point out another important characteristic of the mentioned species, which is related to their chemical and nutritional quality. The species of the genus *Silphium* are characterized by a high content of nutrients, but considering the variability of the available data, especially in relation to the species of the genus *Silphium*, the data on the nutrient composition of the most pronounced species *S. perfoliatum*, *S. trifoliatum*, and *S. integrifolium* should be analyzed in more detail. Kowalska et al. [10] presented various phytonutrients present in the species of the genus *Silphium*, namely, proteins, amino acids (aspartic acid, glutamic acid, and leucine), water-soluble sugars, and minerals (K, Ca, Mg, Fe, and Mn), but also valuable sources of biologically active compounds, mainly ascorbic acid, polyphenols (flavonoids and phenolic acids), terpenoids, oleanosides, etc. [11,12]. In view of this, the cup plant species can be highlighted as an important nutrient source. The situation is quite different with Virginia mallow, about which nutrient potential-based information is still lacking. From the available literature, the genus *Sida* includes more than 200 species [13]—some of which are used in traditional medicine, especially in India (*Ayurveda*), where the roots of *Sida* L. plants are usually used for their pharmacological potential, including their anti-tumor, anti-HIV, and hepatoprotective properties [14]. However, there is still no comprehensive overview of the specific phytochemical, toxicological, and pharmacological properties of the genus *Sida*, especially in relation to the species *Sida hermaphrodita* L [15].

Considering the above-mentioned energetic and nutritional potentials of the species, the time of harvest, i.e., the phenophase of the plant depending on the final purpose, biofuel production, or functional value, is crucial. Both plant species can be harvested once or multiple times a year in temperate (continental) climates. For the cup plant, if the biomass is to be used for biogas production, harvesting typically occurs when the plants are in their full-flowering stage. In addition, the cup plant is harvested for direct combustion after flowering (before seed formation). This is when its biomass moisture content is lowest and when its dry matter yield per unit area is highest. Virginia mallow has peculiarities regarding the occurrence of phenophases, characteristically having an extremely long flowering period, which is why the same plant can have several phenophases, i.e., the formation of buds and up to full flowering. Under these circumstances, harvesting for both biofuel production and direct combustion usually takes place before the first frost.

Based on all of this, and in the absence of published research, the objective of this work was to determine the nutritional, functional, and energy potential of Virginia mal-low and cup plant species in three different phenophases.

## 2. Results and Discussion

### 2.1. Dry Matter Content

An analysis of variance (ANOVA) showed that the dry matter content (DM) significantly differs in relation to the phenophase in both analyzed plant species (Figure 1). The DM content in the leaves of Virginia mallow was in the range of 32.01–35.92%, while for the cup plant the DM values were lower, ranging from 24.28–25.69%. In Virginia mallow, the DM content did not differ within pre- (Ph1) and full-flowering (Ph2) phenophases, with an average value of 32.47%. Generally, the DM content at the end of flowering (Ph3) was about 10% higher than in the pre- (Ph1) and full-flowering phenophase (Ph2). In the cup plant, DM content values did not follow the trend as in Virginia mallow, while in the pre-flowering phase (Ph1) the highest DM was recorded. In addition, the DM values significantly differ in all three analyzed phenophases, with the lowest value recorded in the full-flowering stage (Ph3). The obtained DM values in the leaves for both observed species are considered a very high, especially in relation to the fresh plant material (such as the edible parts, namely, the leaves, fruits, vegetables, etc., of other plant species), which is highly important importance in terms of nutritional potential [16]. A higher DM value shows greater potential for the content of phytochemicals, considering i.e., bioactive compounds (vitamins, minerals, polyphenolics, and so on). The dry matter content in plant tissues is generally strongly dependent on the environmental conditions, specifically, the precipitation, temperature, and air humidity [17], but also on pedological properties (soil type and altitude), genetic characteristics, specific agronomic measures (primarily fertilization), and the plant’s development and phenophase [10,18,19]. As mentioned above, regardless of the phenophase observed, both plant species were found to have high DM content, which can be explained by the climatic conditions that prevailed during the harvest period. Climatological data (Figure 2) show that average daily temperatures ranged from 22 °C to a maximum of 26 °C (in July), depending on the months observed, while the average precipitation (mm) ranged from 20 mm (August) to 259 mm (September) in the same months. That is, high air temperatures and low precipitation resulted in lower water accumulation in the plant tissues, i.e., a higher total dry matter content. Kowalska et al. [10] studied the DM content in *S. perfoliatum* leaves during five phenophases and found an average DM content in the leaves of 17.6%, which is lower compared to the values obtained in this study. In the aforementioned study, the DM content increased steadily as the harvest period progressed, with the lowest content found at the flower bud stage and the highest at the end of the flowering period. In this study, the above trend was found for the Virginia mallow, but for the cup plant there was a deviation observed, with the highest DM content in the pre-flowering stage and the lowest in the full-flowering stage.

### 2.2. Energetic Characteristics of Virginia Mallow and Cup Plant

A proximate analysis (Table 1) includes several valid studied parameters that can be used to determine the quality of raw material for biofuel production. In direct combustion, it is desirable to have the lowest possible ash content since ash has no calorific value and thus reduces the efficiency of the combustion system [20]. The ash content in biomass can be up to 40%, and if the biomass has an ash content lower than 1%, then it is considered to be of the best quality [21]. In this study, the ash content was 3% for Virginia mallow and about 6% for the cup plant. Unlike ash, coke is considered to be an extremely important fuel property, and for biomass it is desirable to have as much of it as possible. The coke content in Virginia mallow was about 10%, while in the cup plant it was almost 16%. Considering other energy crops, for example, Miscanthus, which is considered the best energy crop and contains about 12% coke [22], it can be said that the crops in this study also have a great potential for direct combustion. Another important indicator of fuel quality is calorific value, which is the amount of energy released when one unit of fuel mass is completely burned, the gases are cooled to 25 °C, and water escapes as condensate [23]. The results of this study for Virginia mallow and cup plants show that the values obtained are in the range of 15–20 MJ/kg, which is an average calorific value for agricultural biomass [24].

The elemental analysis of Virginia mallow and cup plants (Table 2) included an analysis of the total carbon (C), hydrogen (H), nitrogen (N), sulfur (S), and oxygen (O). The most important element in biomass is carbon (C), which occurs in organic compounds in the plant, combines with oxygen, and generates a large amount of heat energy [25]. Agricultural biomass contains on average about 48% carbon, and the results obtained in this study are slightly higher (for Virginia mallow, almost 50%), which shows a high heat energy potential. Sulfur is an element that should be present in biomass in the lowest possible amounts, preferably in the range of 0.02–0.23%, as it is directly associated with reducing CO_2_ emissions by up to 75% [26]. In fact, during the conversion of biomass to energy, sulfur is involved in the formation of sulfur dioxide (SO_2_), which later leads to the formation of acid rain, one of the most dangerous environmental and ecosystem pollutants, and can significantly affect crop production [23]. Low levels of sulfur were observed in Virginia mallow (0.06%) and the cup plant (0.02%), which is an indicator of the potential of the mentioned plant species for direct combustion. Hydrogen, along with nitrogen, forms the basic fuel composition of any fuel, and an increased hydrogen content improves fuel quality by positively affecting the oxygen levels. Agricultural biomass contains on average about 6% hydrogen [21,26,27], which is in line with the results of this research.

### 2.3. Specialized Metabolites of Fresh Leaves

The content of the analyzed specialized metabolites (SM) in the fresh leaf samples of Virginia mallow and the cup plant differed significantly depending on the phenophase. The ascorbic acid content (AsA) determined in the leaves of Virginia mallow did not differ significantly in the pre- (Ph1) and full- (Ph2) flowering phenophases with an average value of 226.43 mg/100 g fw. At the end of the flowering (Ph3) phase, the AsA content significantly decreased; even a 33% lower value was recorded compared to the Ph1 and Ph2 stages. The obtained results are expected due to the fact that at the beginning of flowering the AsA levels decrease significantly since the flowering process in plants is delayed by high levels of AsA [28]. The opposite trend regarding the phenophase was determined for the AsA content in the leaves of the cup plant depending on the plant phenophase. The highest AsA content was recorded at the end of flowering with an approximately four times higher value compared to the pre- and full-flowering stages, which did not statistically differ, with an average AsA content of 31.23 mg/100 g fw. A similar trend regarding the AsA content in the *S. perfoliatum* leaves depending on the phenophase of the plant was observed in the study by Kowalska et al. [10], in which the AsA content was higher after spring growth, decreased significantly in the flower bud phase, and showed an increasing trend in the other phenophases, i.e., at the beginning, full-flowering stage, and the end of flowering. In the phenophase of the end of flowering, nearly the same AsA values were reached as in the first phase, spring regrowth. Certain biotic and abiotic conditions to which plants are exposed during their growth and development can significantly affect their content of bioactive compounds. For example, drought, as a form of plant stress, can contribute to a plant synthesizing more bioactive compounds for protection against or adaptation to external conditions [29]. So, it can be assumed that the increase in the ascorbic acid content in the cup plant during the end of flowering occurred as a plant response to extreme climatic conditions, namely, above-average high temperatures and low rainfall (dry period of three months). For plants to survive the negative effects of environmental stress, they must adjust their metabolic and physiological processes responsible for normal organismal functions, such as protein synthesis, enzymatic activity, and photosynthetic capacity [30,31]. The importance of AsA in plants is also demonstrated by research showing that an exogenous administration of AsA is an effective way to protect plants from abiotic stress [32]. Ascorbic acid is not only actively involved in plant metabolism but also in the detoxification of reactive oxygen species (ROS), whose increase leads to a decrease in photosynthetic activity, which in turn results in accelerated plant senescence [33].

Besides ascorbic acid, polyphenols are also SM that occur in plant metabolism as potent antioxidants with beneficial effects on human health [34,35]. In this study, the content of total phenolics (TPC), flavonoids (TFC), and non-flavonoids (NTFC) were determined, which varied significantly according to the phenophase of the observed species, as shown in Table 3. Virginia mallow contained up to 30% more TPC at the pre-flowering (Ph1) stage than at full flowering (Ph2), while at the end of flowering (Ph3) the TPC returned to almost the same amount as in Ph1. In contrast to Virginia mallow, the cup plant produced the highest concentration of phenolics in the phase of full flowering (Ph2)—up to 60% more than in the pre-flowering phase (Ph1). In plants, phenolics are involved in growth control, flavor formation, and insect attraction functions, as well as in protection against biotic and abiotic factors. Some studies have also found an increase in the TPC from the budding to flowering stages, which was also observed in this study, wherein the cup plants showed a value of 65%. The reason for the increase in the total phenol content may be the plant’s response to the failed redistribution of phenol in the developing plant parts or part of a defense mechanism against senescence with which the plant responds by producing more phenol [36].

The high values for the total flavonoid (TFC) and total non-flavonoid content (TNFC) were also determined in the leaves of both the Virginia mallow and cup plant. Considering the values obtained, it can be pointed out that the specific phenophase significantly affects the TFC and TNFC. The highest TFC values in the leaves of Virginia mallow were recorded at the end of the flowering period (Ph3): they were up 25% compared to the Ph1 stage. In the cup plant, the highest TFC was determined in the full-flowering stage (Ph2)—40% more than in the pre-flowering stage (Ph1). It is interesting to note that the flavonoid content showed a negative trend at the end of flowering (Ph3) compared to full flowering (Ph2), possibly due to the agro-climatic conditions, i.e., the end of the dry days that prevailed during the Ph2 phase. The content of non-flavonoids (TNFC) in the leaves of the cup plant where the highest in the full-flowering (Ph2) phase and were about 40% higher compared to the pre-flowering phase (Ph1). Even more interesting results were found in Virginia mallow, where the highest TNFC was obtained in the Ph1 phase, up by almost 50% compared to the Ph2 phase (Table 3). Plants stimulate the immune system as part of their defense mechanism, with a tendency to increase the synthesis of phenols for the detoxification of accumulated ROS, which is a direct consequence of environmental stress conditions [37]. These results indicate that the studied species represent a potential and extremely valuable food source for further use.

Besides the total polyphenolic compounds, some of the individual phenolics were determined and quantified (Table 4). From the group of hydroxycinnamic acid, caffeic, coumaric, and ferulic acids were determined; from the ellagitannins ellagic acid and from the flavanone-7-O-glycoside, naringin was recorded. The caffeic acid content in Virginia mallow was in the range from 13.07–32.9 mg/L. Higher amounts of caffeic acid were determined in the pre- (Ph1) and full-flowering (Ph2) phenophases with no recorded statistical difference between them (average value of 29.48 mg/L). At the end of flowering (Ph3), the amount of caffeic acid decreased more than twofold. An identical situation in Virginia mallow leaves was found for ellagic acid, which averaged 1400 mg/L in Ph1 and Ph2 and decreased by almost 20% in Ph3. In contrast to caffeic acid and ellagic acid, the concentration of coumaric acid was highest in the Ph3 phase, at 30.55 mg/L, which is 75% more than in the Ph1 and Ph2 phenophases. No statistical difference was found regarding the ferulic acid content in the leaves of Virginia mallow between the phenophases with an average value of 4.37 mg/L. the naringin content differed significantly in all three phenophases observed, with the lowest content determined in Ph3 at 530.35 mg/L, while in Ph1 the determined value was 2.5 times higher, showing the tendency of a decrease with the aging (senescence) of the plant.

In the leaves of the cup plant, the content of caffeic acid was highest at the full-flowering stage (Ph2), and about 15% lower at the pre-flowering stage (Ph1), while the content of caffeic acid in the leaves at the end of flowering (Ph3) was not determined. An opposite trend was observed for coumaric acid as a function of the phenophase, i.e., it was not determined in Ph1 and Ph2, while in Ph3 (end of flowering) as much as 287.43 mg/L was measured. In the case of ellagic acid, there was an increasing trend with respect to the phenophase, meaning that a higher content was synthesized as the plant aged. For example, in the Ph1 phase, the amount of ellagic acid was 46.63 mg/L, while the content increased to about 40% in the Ph2 and even 70% in Ph3. The lowest content of ferulic acid was observed in the Ph1 phase, at 28.43 mg/L, while in the full-flowering stage (Ph2) an increase of about 80% was observed; in the Ph3 phase, a decrease in the content was again confirmed, but in the mentioned phenophase it was still slightly higher than in the pre-flowering phase (Ph1). The naringin content in the cup plant was not significantly different in Ph1 and Ph3 with an average of 145.88 mg/L, but it increased twofold in the Ph2 phenophase. Kowalska and Wolski [13] also determined a broad spectrum of individual polyphenols in *S. perfoliatum* L. depending on the plant organ; especially in the leaves, caffeic acid and p-coumaric acid were determined as predominant from the hydroxycinnamic acids, proto-catechuic acid, p-hydroxybenzoic acid, and vanillic acid from the hydroxybenzoic acids.

Hydroxycinnamic acids and their derivatives, as well as ellagitannins and their derivatives, are characterized by beneficial medicinal properties towards human health. They are mainly characterized by their potent antioxidant, anti-collagenase, anti-inflammatory, antimicrobial, and anti-tyrosinase activities, as well as their UV-protective effects, suggesting that they can be used as anti-aging and anti-inflammatory agents [38,39,40,41]. Considering that most of the mentioned polyphenolic compounds are specific to fruits, vegetables, and herbs and their products, it is important to point out that the leaves of the species *Sida hermaphrodita* L. and *Silphium perfoliatum* L., given the relatively high contents of the individual polyphenolic compounds analyzed in this study, represent a very valuable source that could benefit the pharmaceutical, food, or cosmetic industries.

### 2.4. Pigment Compounds Content

Pigment compounds, including chlorophylls and carotenoids, are important phytochemicals that are not only necessary for the normal development of the physiological functions of plants, but also excel as potent antioxidants with numerous beneficial biological functions and health benefits [42,43]. In the early stages of development, the chlorophyll content increases, but with senescence, the chlorophyll content decreases, and other pigments accumulate. In this research, from the pigments chlorophyll a (Chl a) and chlorophyll b (Chl b), the total chlorophylls (TCh) and total carotenoids (TCa) were quantified in the leaves of Virginia mallow and the cup plant (Table 5).

In leaves of Virginia mallow, the content of Chl a at the pre- (Ph1) and full-flowering (Ph2) phases did not differ statistically, averaging 0.97 mg/g. As the plant aged, i.e., at the end of flowering (Ph3), the amount of Chl a decreased by 30% compared to Ph1 and Ph2. In the case of Chl b, the highest content of pigment was synthesized in the Ph2 phase—0.65 mg/g. In contrast to the Ph2 phenophase, the amount of Chl b was 40% lower in the Ph1 phenophase, which is also the lowest value measured (0.47 mg/g). Although a higher Chl b content was measured in the Ph3 phase than in the Ph1 phase, it was still 20% lower than in the full-flowering phase (Ph2). The total chlorophyll content of Virginia mallow did not differ statistically between Ph1 and Ph3, averaging 1.37 mg/g, while in Ph2 it reached a maximum of 1.64 mg/g, which was a 20% increase. With the delay of the phenophase of the plant, i.e., senescence, as hypothesized, the decrease in the total chlorophyll content led to an increase in the total carotenoid content, which was also confirmed according to the results for Virginia mallow. The content of total carotenoids was lowest in the Ph1 and Ph2 phases, while in the Ph3 phenophase (end of flowering) the value of total carotenoids was almost 45% higher than in the previous two phenophases. The Cup plant yielded statistically very similar values for all observed pigment compounds in the pre-flowering (Ph1) and full-flowering (Ph2) phenophases, while at the end of the flowering (Ph3) phase, significantly lower values were observed. For example, the content of chlorophyll a was 2-fold lower and the content of chlorophyll b was as much as 2.5-fold lower in the end of flowering phase (Ph3) than in the earlier phenophases. The same trend continued with respect to the total chlorophyll content, where the value at the end of flowering (Ph3) decreased more than twofold. It is interesting to note that carotenoids also showed a decrease in value towards the end of flowering, but not as severe, such that the total carotenoid content in Ph3 was 0.09 mg/g, which is about 70% lower than in the flowering phase.

### 2.5. Antioxidant Capacity

The antioxidant capacity of plants is closely related to the content of specialized metabolites, i.e., bioactive compounds. Since significantly high levels of polyphenols, ascorbic acid, chlorophylls, and carotenoids were found in the leaves of the studied species, namely, Virginia mallow and the cup plant, high values of antioxidant capacity were expected and confirmed (Figure 3 and Figure 4).

Considering the values of the antioxidant capacity determined for Virginia mallow (Figure 3), we note that there is no statistical difference between the phase of full flowering (Ph2) and the end of flowering (Ph3) with an average antioxidant capacity of 4605.58 µmol TE/L according to the FRAP method. A significantly higher value was observed in the phase before flowering (Ph1), proving that the antioxidant capacity of the plant decreases during the transition from vegetative growth to generative development; the same trend was observed for phenolic compounds. Using the ABTS method, opposite results were observed concerning the phenophase of the plant, in which the lowest antioxidant capacity was determined in Ph1 and Ph2 without statistical difference, while the highest value was obtained in Ph3 with 2511.60 µmol TE/L. The results regarding the antioxidant capacity of the fresh leaves of the cup plant are shown in Figure 4. From the values obtained, we can conclude that the phenophase of plant development significantly affects the antioxidant capacity and that the highest value was measured by the FRAP method at the end of the subsidence phase (Ph3) (4810.06 µmol TE/L) and the lowest in Ph1 (4540.92 µmol TE/L). On the other hand, the ABTS method found that phenophase had no statistically significant effect on antioxidant capacity. These results can be explained by the sensitivity of the method, because the ABTS and FRAP methods cover a wide range of polyphenolic compounds, including hydrophilic and lipophilic antioxidant systems. Therefore, it is always desirable to combine at least two methods to gain insights into both hydrophilic and lipophilic systems. It is worth noting that regardless of the phenophase and combined method, very high antioxidant capacity values were obtained for both Virginia mallow and cup plant, highlighting the leaves of these species as potent and promising antioxidants.

## 3. Materials and Methods

### 3.1. Plant Material

Virginia mallow (*Sida hermaphrodita* L.) and cup plant (*Silphium perfoliatum* L.) were grown in the experimental fields of the University of Zagreb, Faculty of Agriculture, Maksimir (elevation 123 m, 45°49′48″ N 16°01′19″ E). The experimental fields were established in May 2017 by planting seedlings. Seedlings of both crops were planted 0.75 m apart within and between rows in three replicates. The base plot had an area of 8.44 m^2^. During the vegetation period, no agricultural measures were carried out, i.e., fertilization or irrigation were not carried out. Biomass samples collected for analysis of energetic properties were taken in late September 2022.

Leaf samples were collected at three different phenophases in 2022 for analysis of specialized metabolites. The first leaf mass samples were collected before flowering (Ph1), then at full-flowering stage (Ph2), and at the end of flowering (Ph3). Regarding the different development stages of the two studied species, the exact dates of the plant samples are shown in Table 6. All analyses of energy and nutritional properties were performed in the laboratory of the Department of Agricultural Technology, Storage, and Transport of the Faculty of Agriculture, University of Zagreb.

During the vegetation period, specifically, from June to September, the basic climatic characteristics of the Maksimir experimental station were monitored, i.e., air temperature (minimum, maximum and average) and precipitation. The data were obtained from the Croatian Meteorological and Hydrological Service [44] and are presented in Figure 2.

### 3.2. Total Dry Matter Content

Total dry matter content (DM, %) was determined by drying at 105 °C to constant weight using the standard laboratory protocol according to AOAC [45] in drying oven. (Memmert model 30-1060, Memmert GmbH + Co. KG, Schwabach, Germany).

### 3.3. Energy Biomass Characterization

All laboratory procedures were performed in three replicates, and data were expressed as means of dry matter. The 2 kg of fresh above-ground biomass was dried in a dryer at 60 °C for 48 h [46]. Before further analysis, samples were coarsely ground using a blender (Retsch Grindomix GM 300, Haan, Germany), crushed in a laboratory mill (IKA Analysentechnik GmbH, Staufen, Germany), and sieved on a sieve-shaker (Retsch AS 200, Retsch GmbH, Haan, Germany). For the laboratory analyses, 150 g of a dry sample with a particle size of 250 μm to 1000 μm was used, which was then analyzed by standard methods. A muffle furnace (Nabertherm GmbH, Nabertherm Controller B170, Lilienthal, Germany) was used to determine the content of coke [47] and ash [48]. A sample of 0.5 g was weighed in empty porcelain vessels, which were then placed in a preheated oven at 900 °C/5 min for coke analysis, followed by cooling to room temperature. The crucibles were weighed when they reached room temperature and the percentage of coke was calculated. To determine the ash content, the procedure followed with respect to the crucibles was identical to the coke analysis, but now the samples were burned for 5 h at a temperature of 550 °C with a steady increase in temperature. The content of volatile substances and fixed carbon was mathematically calculated from the difference. The content of nitrogen, hydrogen, carbon [49], and sulfur [50] was determined by dry combustion in a Vario Macro CHNS analyzer using He gas to transport the sample (Elementar Analysensysteme GmbH, Langenselbold, Germany). An amount of 150 mg of tungsten oxide was weighed on aluminum foil (WO_3_) in powder form, which serves as a catalyst, and then another 50 mg was added to the dry sample. The aluminum foil was then crimpled and pressed to form a small ball, which was placed in the autosampler. The oxygen content was determined mathematically from the difference of the other elements determined.

A higher heating value [51] was determined at IKA C200 Analysentechnik GmbH-Heitersheim (Staufen, Germany), and a lower heating value was calculated mathematically using the ISO method. A total of 0.2 g of the dry sample was weighed in a quartz crucible, which was placed in a calorimetric bomb that was then placed in the instrument.

### 3.4. Determination of Specialized Metabolites

To determine the amount of ascorbic acid (AsA), 10 g of the sample was weighed and quantitatively transferred to a 100 mL volumetric flask with a 2% oxalic acid solution. Then, the contents of the flask were filtered, and the resulting filtrate was used for further analysis. A volume of 10 mL of the resulting filtrate was pipetted into a 50 mL Erlenmeyer flask and then titrated with a solution of 2,6 dichlorindolphenol (DCPIP) until a pink, stable coloration of solution appeared. The amount of AsA was expressed in mg/100 g fresh weight.

Total phenolics (TPC), flavonoid (TFC), and non-flavonoid (TNFC) content was determined spectrophotometrically (Shimadzu, UV 1900i, Duisburg, Germany) at 750 nm according to the method described by Ough and Amerine [52]. The method is based on the color reaction that phenols develop using the Folin–Ciocalteu reagent. The sample is prepared by weighing 10 g ± 0.01 of fresh leaves and transferring them to an Erlenmeyer flask, into which the first 40 mL of 80% EtOH (*v*/*v*) is poured. The sample thus prepared was heated to the boiling point, and it was then kept under reflux for an additional 10 min. After 10 min, the sample was filtered through Whatman filter paper into a 100 mL volumetric flask. After filtration, the remainder of the sample was transferred to an Erlenmeyer flask, into which another 50 mL of 80% EtOH (*v*/*v*) was added, and the procedure was repeated under reflux for 10 min. The sample was filtered, the filtrates were combined, and the flask was made up to the mark with 80% EtOH (*v*/*v*). The reaction with Folin–Ciocalteu reagent was prepared in the following order: 0.5 mL of the ethanolic extract was added to a 50 mL volumetric flask, followed by 30 mL distilled water (dH_2_O), 2.5 mL of the prepared Folin–Ciocalteu reagent (1:2 with dH_2_O), and 7.5 mL of saturated sodium carbonate solution (Na_2_CO_3_); the flask was filled to the mark with dH_2_O and the prepared sample was allowed to stand for 2 h at room temperature with occasional shaking. The same ethanol extracts were used for TFC and TNFC. The determination of TNFC was performed according to the following procedure: a total of 10 mL of the ethanolic extract was transferred into a flask with a volume of 25 mL, into which 5 mL of HCl (1:4, *v*/*v*) and 5 mL of formaldehyde were then added. The obtained solution was then blown with nitrogen (N_2_) and the samples were allowed to stand for 24 h in a dark room at room temperature. After 24 h, the samples were filtered through Whatman filter paper and the same Folin–Ciocalteu reaction was performed as for TPC. Absorbance of blue solutions was measured spectrophotometrically at 750 nm using distilled water as a blank. Gallic acid and catechol were used as external standards, and the TPC and TNFC were expressed as mg GAE/100 g fresh weight (fw). TFC was mathematically calculated and expressed as the difference between total phenols and non-flavonoids.

According to the method of Holm [53] and Wattstein [54], chlorophyll a (Chl_a), chlorophyll b (Chl_b), total chlorophylls (TCh), and total carotenoids (TCA) were determined. The process of extraction and determination of pigments included weighing 0.2 ± 0.01 g of a fresh leaf sample onto which 15 mL of acetone (p.a.) was added, followed by homogenization with a laboratory homogenizer (IKA, UltraTurax T-18, Staufencity, Germany). The homogenization procedure was repeated two more times and 15 mL of acetone was added each time. Following three additions of acetone, the solution was filtered through Whatman filter paper and transferred to a 25 mL volumetric flask. Absorbance was measured spectrophotometrically (Shimadzu UV 1900i, Duisburg, Germany) at 662, 644, and 440 nm using acetone as a blank. Holm–Wettstein Equations (1) were used to quantify each pigment and the final content was expressed in mg/g.
(1)$$Chl\_a=9.784\;\times \;A_{662}-0.990\;\times \;A_{662}\;Chl_{b}=21.426\;\times \;A_{664}-4.65\;\times \;A_{622}\;TCh=5.134\;\times \;A_{662}+20.436\;\times \;A_{644}\;TCA=4.695\;\times \;A_{440}-0.268\;\times \;TCh\;\left[mg/L\right]$$

### 3.5. High-Performance Liquid Chromatography (HPLC)

High-performance liquid chromatography (HPLC) was used to analyze the extracted individual phenolic compounds according to the modified method previously described by Otles and Yalcin [55]. Amounts of 1 g ± 0.01 of fresh leaves of Virginia mallow and cup plant were homogenized with 10 mL of 80% methanol (*v*/*v*) using a laboratory homogenizer (IKA, UltraTurax T-18, Staufencity, Germany). The mixtures were then further homogenized for 30 min at 50 °C in closed beakers in an ultrasonic bath (Bandelin RK 103H, Berlin, Germany). The solutions were then filtered through Whatman filter paper, and the obtained eluates were filtered again through Chromafil PA filters before injection into the vials. Separation, identification, and quantification of polyphenols were performed by HPLC analysis using LC Nexera (Shimadzu, Kyoto, Japan) equipped with a photodiode array and fluorescent detector (PDA-RF). The parameters of chromatographic analysis were defined according to Repajić et al. [56] with minor modifications. Separation was carried out on a NUCLEOSIL 100-5 C18 (5 µm, 250 × 4.6 mm i.d.) column (Macherey-Nagel, GmbH, Düren, Germany). The flow rate was 0.9 mL/min, and the applied volume of the samples was 20 μL. For gradient elution, mobile phase A contained 3% formic acid in HPLC-grade water (*v*/*v*), while mobile phase B contained 3% formic acid in acetonitrile (*v*/*v*). The analyses were performed at a temperature of 25 °C, and the duration of the single run was 45 min for each sample. Detection of phenols was performed with UV/VIS–PDA detector at wavelengths ranging from 220 to 360 nm. The phenolic compounds were identified based on their retention times compared to standards purchased from Sigma Aldrich (Steinheim, Germany). For quantitative analyzes, calibration curves (Table 7) were obtained by injecting the mixed standard solution with different concentrations (2, 10, 40, and 100 μg/mL).

All determinations were performed in triplicate and the results for individual phenolic compounds (caffeic acid, coumaric acid, ellagic acid, ferulic acid, and naringin) were expressed as mg/L. The chromatograms of the HPLC analysis of the individual polyphenolic compounds corresponding to the plant phenophase are shown for both species in Figure 5 and Figure 6.

### 3.6. Determination of Antioxidant Capacity (FRAP and ABTS)

Antioxidant capacity was determined by the method of ferric reducing antioxidant power (FRAP) according to the research of Benzie and Strain [57]. The method is based on the reduction of Fe^3+^ (ferrous) ions to Fe^2+^ (ferrous) ions, the latter forming an intense blue complex with tripyridyltriazine (TPTZ = 2,4,6-tripyridyl-S-triazine), Fe^2+^/TPTZ, at very low pH values. The FRAP reagent was freshly prepared by mixing acetate buffer (0.3 M), TPTZ reagent (10 mM) and ferric chloride (FeCl_3_ x 6H_2_O; 20 mM) in a 10:1:1 ratio. The reaction mixtures contained: 720 µL dH_2_O, 240 µL extracts, and 6240 µL FRAP reagent. For the blank, 80% EtOH (*v*/*v*) was used instead of the sample extract. Following the method, the incubation of the prepared mixtures was performed at 37 °C, while absorbance was measured spectrophotometrically (Shimadzu, 1900i, Kyoto, Japan) after 5 min at 593 nm. Trolox (6-hydroxy-2,5,7,8-tetramethylchroman-2-carboxylic acid, Sigma-Aldrich, St. Louis, MO, USA) was used as the antioxidant standard (Table 8) and results were expressed as µmol TE/L (Trolox equivalents).

Antioxidant capacity was also determined by the ABTS test, which was described by Miller et al. [57], using the cationic radical 2,2′-azinobis (3-ethylbenzothiazoline-6-sulfonic acid). In the presence of antioxidants, ABTS+ radicals are reduced to ABTS, and the reaction is shown by a reduction in color from pale green to transparent. To prepare a stable ABTS+ solution, 88 µL of a 140 mM potassium persulfate solution was taken, mixed with 5 mL of a 7 mM ABTS solution, and allowed to stand in the dark at room temperature for 14–16 h. On the day of analysis, a 1% solution of ABTS in 96% ethanol was prepared and the absorbance was measured at 734 nm. Later, 160 µL of the extract mixed with 2 mL of the 1% ABTS+ solution was poured directly into the cuvette, and after 5 min, the absorbance was measured spectrophotometrically (Shimadzu, 1900i, Kyoto, Japan) at 734 nm. For the blank, 96% ethanol was used, while Trolox was used as an antioxidant standard. Final results were calculated based on the calibration curve (Table 8) and expressed as µmol TE/L.

### 3.7. Statistical Analysis

The data were studied by applying the statistical program SAS version 9.4. [58]. All measurements were performed in three repetitions, from which the mean value was calculated. The deviation of the results from the mean is indicated by the standard deviation (SD), listed in the same row of the table, and marked with a different upper- and lower-case letter (^a–c^). Means were compared using the least significant difference (LSD) t-test and considered significantly different at *p* ≤ 0.01. In addition to the results, different letters denoted by the standard deviation value indicate a statistically significant difference between the observed phenophases at *p* ≤ 0.0001.

## 4. Conclusions

From the obtained results, it can be concluded that Virginia mallow and the cup plant are good sources for producing “green energy”. They have extremely favorable biomass quality characteristics, high coke and carbon content, and high calorific value for use in biogas production or direct combustion. On the other hand, the leaves of the studied plants showed a high nutritional potential, which was significantly influenced by the phenophase. High levels of ascorbic acid, polyphenols (mainly hydroxycinnamic acids and ellagitannins), chlorophylls, and carotenoids, as well as a high antioxidant capacity, were found in both plants. As mentioned above, the phenophase of the plants has a strong influence on the content of the analyzed metabolites. Thus, the highest levels of bioactive compounds in the leaves were found at full flowering and at the end of the flowering period. In addition to their significant energy potential, Virginia mallow and the cup plant have high nutritional and functional potentials and can be considered as a new value-added by-product. Due to their high nutritional value, the leaves of the studied species can be used for the production of various by-products in the pharmaceutical, cosmetic, and food industries, while the rest of the plant biomass can be used for energy production, thereby realizing a zero-waste concept.

## Figures and Tables

**Figure 1 plants-11-02906-f001:**
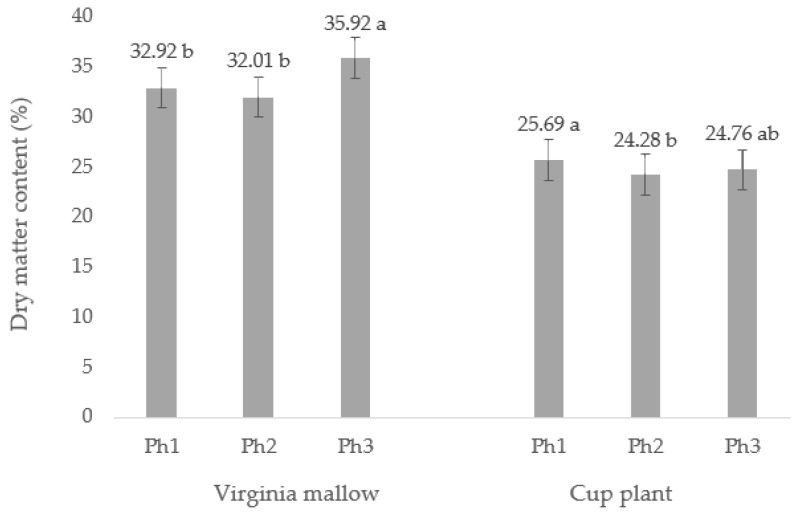
Dry matter content (%) of leaves of Virginia mallow and cup plant at different phenophase. Ph1—pre-flowering, Ph2—full flowering, and Ph3—end of flowering. Different letters indicate significant differences between means at *p* ≤ 0.0001.

**Figure 2 plants-11-02906-f002:**
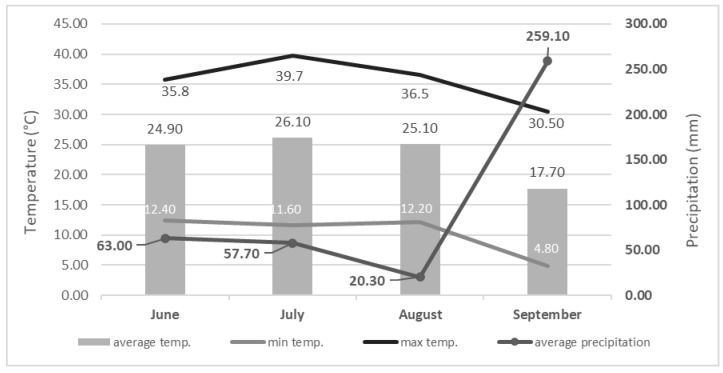
Climate diagram with data from the meteorological station ‘Maksimir’ for the period of June (6) to September (9) (Croatian Meteorological and Hydrological Service, 2022).

**Figure 3 plants-11-02906-f003:**
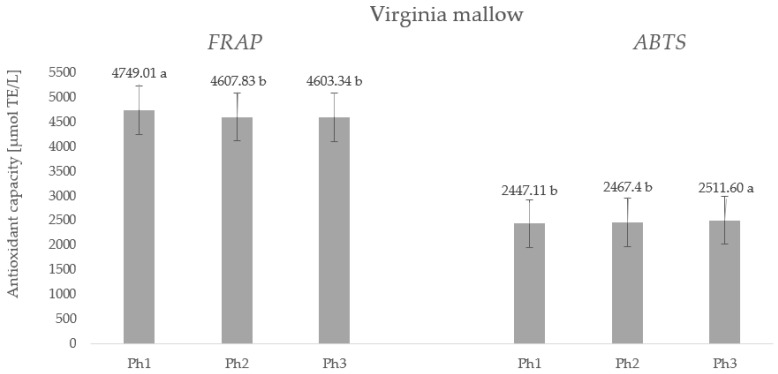
Antioxidant capacity (µmol TE/L) of fresh leaf of Virginia mallow. Results are expressed as mean ± standard deviation. Different letters indicate significant differences between mean values.

**Figure 4 plants-11-02906-f004:**
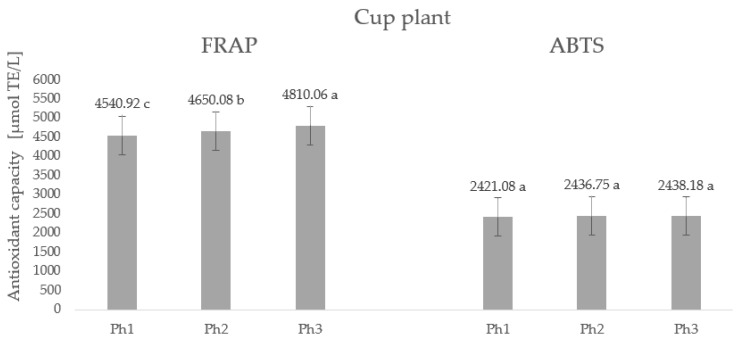
Antioxidant capacity (µmol TE/L) of fresh leaves of cup plant. Results are expressed as mean ± standard deviation. Different letters indicate significant differences between mean values.

**Figure 5 plants-11-02906-f005:**
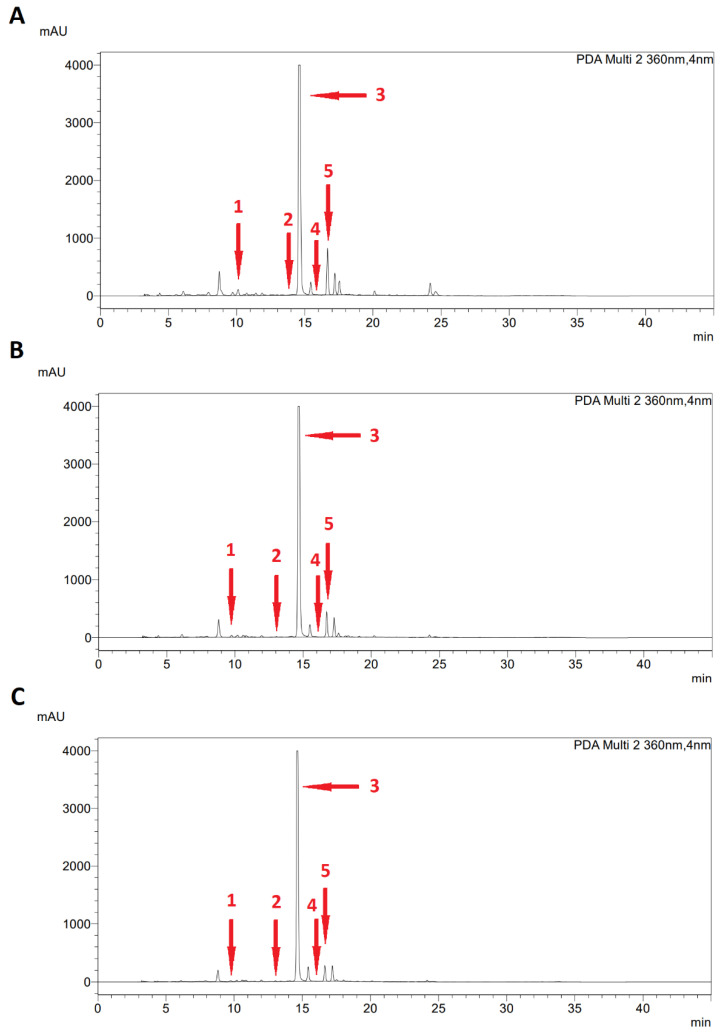
HPLC chromatogram of phenolic compounds profile of *Sida hermaphrodita* L. recorded at 360 nm, at (**A**) pre-flowering, (**B**) flowering, and (**C**) end of flowering stage; 1—caffeic acid, 2—coumaric acid, 3—ellagic acid, 4—ferulic acid, and 5—naringin.

**Figure 6 plants-11-02906-f006:**
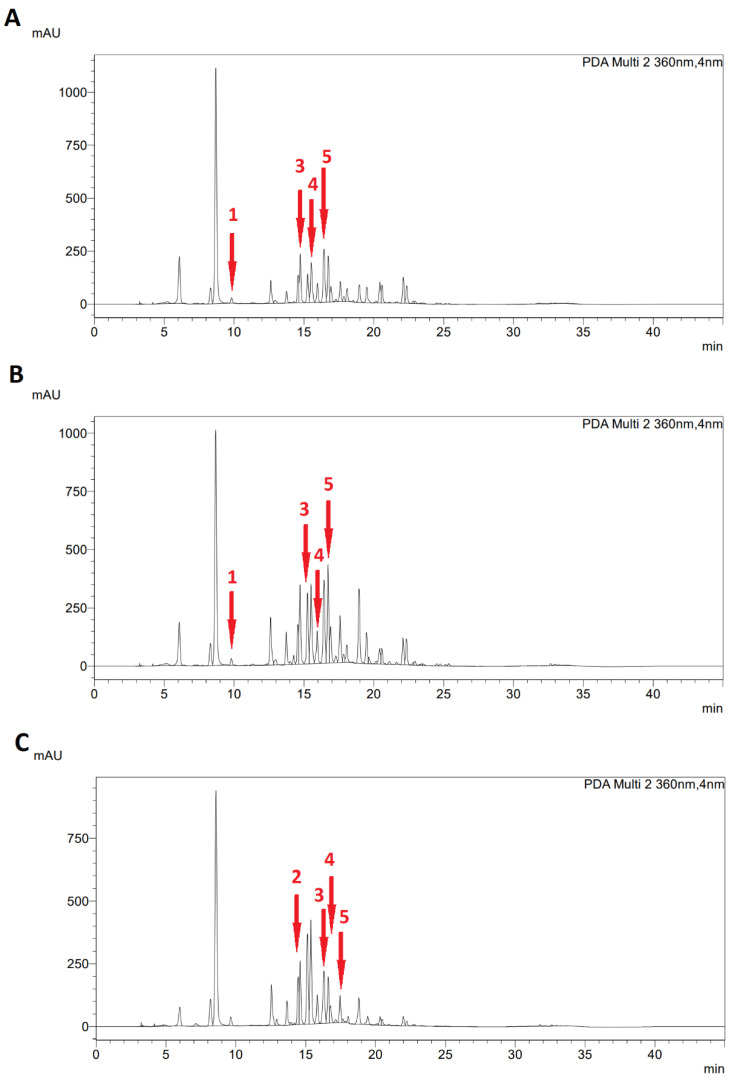
HPLC chromatogram of phenolic compounds profile of *Sylphium perfoliatum* L. at (**A**) pre-flowering, (**B**) flowering, and (**C**) end of flowering stage; 1—caffeic acid, 2—coumaric acid, 3—ellagic acid, 4—ferulic acid, and 5—naringin.

**Table 1 plants-11-02906-t001:** Proximate analysis of Virginia mallow and cup plant biomasses.

	Virginia Mallow	Cup Plant
Ash (%)	2.92	5.86
Coke (%)	10.45	15.49
Fixed Carbon (%)	7.53	9.28
Volatile Matters (%)	80.31	74.45
HHV * (MJ/kg)	17.36	15.46
LHV * (MJ/kg)	16.02	14.54

* HHV—higher heating value; LHV—lower heating value.

**Table 2 plants-11-02906-t002:** Elemental analysis of Virginia mallow and cup plant biomass.

	Virginia Mallow	Cup Plant
Carbon (C)	49.57%	42.94%
Hydrogen (H)	6.15%	5.30%
Nitrogen (N)	0.60%	0.37%
Sulfur (S)	0.06%	0.02%
Oxygen (O)	43.63%	50.92%

**Table 3 plants-11-02906-t003:** Specialized metabolites content of Virginia mallow and cup plant leaves in different phenophases.

Phenophases	AsA (mg/100 g fw)	TPC (mg GAE/100 g fw)	TFC (mg GAE/100 g fw)	TNFC (mg GAE/100 g fw)
**Virginia Mallow**				
Ph_1_	223.07 ± 10.17 ^a^	1079.59 ± 0.60 ^a^	434.00 ± 0.83 ^b^	645.59 ± 1.16 ^a^
Ph_2_	229.79 ± 11.09 ^a^	835.58 ± 1.33 ^c^	393.44 ± 2.21 ^c^	442.15 ± 1.82 ^b^
Ph_3_	170.30 ± 2.36 ^b^	1072.45 ± 0.92 ^b^	493.13 ± 1.90 ^a^	579.32 ± 2.81 ^c^
LSD	26.624	3.021	5.305	6.197
ANOVA	*p* ≤ 0.0003	*p* ≤ 0.0001	*p* ≤ 0.0001	*p* ≤ 0.0001
**Cup plant**				
Ph_1_	24.42 ± 1.95 ^b^	674.94 ± 0.15 ^c^	399.09 ± 2.12 ^c^	399.09 ± 2.21 ^c^
Ph_2_	38.10 ± 5.19 ^b^	1115.21 ± 0.59 ^a^	557.25 ± 1.70 ^a^	557.25 ± 2.00 ^a^
Ph_3_	122.57 ± 9.36 ^a^	917.88 ± 2.45 ^b^	492.67 ± 4.25 ^b^	492.67 ± 1.91 ^b^
LSD	19.007	4.422	6.190	6.190
ANOVA	*p* ≤ 0.0001	*p* ≤ 0.0001	*p* ≤ 0.0001	*p* ≤ 0.0001

Ph1—pre-flowering, Ph2—full flowering, and Ph3—end of flowering; AsA—ascorbic acid content; TPC—total phenol content; TFC—total flavonoid content; TNFC—total non-flavonoid content. Results are expressed as mean ± standard deviation. Different letters indicate significant differences between mean values at *p* ≤ 0.0001.

**Table 4 plants-11-02906-t004:** Individual polyphenolics (mg/L) in leaves of Virginia mallow and cup plant leaves in different phenophases.

Phenophases	Caffeic Acid	Coumaric Acid	Ellagic Acid	Ferulic Acid	Naringin
**Virginia Mallow**					
Ph_1_	32.90 ± 6.46 ^a^	15.25 ± 2.48 ^b^	1421.95 ± 75.43 ^a^	4.27 ± 0.02	1404.98 ± 120.94 ^a^
Ph_2_	26.05 ± 0.64 ^a^	19.90 ± 1.35 ^b^	1398.43 ± 13.23 ^a^	4.74 ± 0.57	760.79 ± 22.84 ^b^
Ph_3_	13.07 ± 0.58 ^b^	30.55 ± 0.45 ^a^	1213.63 ± 16.58 ^b^	4.10 ± 0.07	530.35 ± 10.81^c^
LSD	11.384	5.0047	136.95	0.978	215.93
ANOVA	*p* ≤ 0.0018	*p* ≤ 0.0001	*p* ≤ 0.0025	NS	*p* ≤ 0.0001
**Cup plant**					
Ph_1_	16.22 ± 0.07 ^b^	0.00 ^b^	46.63 ± 3.94 ^c^	28.43 ± 0.71 ^c^	157.07 ± 2.98 ^b^
Ph_2_	18.62 ± 0.24 ^a^	0.00 ^b^	66.71 ± 5.36 ^b^	50.77 ± 3.28 ^a^	302.57 ± 15.42 ^a^
Ph_3_	0.00^c^	287.43 ± 0.01 ^a^	81.95 ± 1.45 ^a^	37.45 ± 0.06 ^b^	134.68 ± 4.12 ^b^
LSD	0.4479	0.0267	11.904	5.8643	28.381
ANOVA	*p* ≤ 0.0001	*p* ≤ 0.0001	*p* ≤ 0.0001	*p* ≤ 0.0001	*p* ≤ 0.0001

Ph1—pre-flowering, Ph2—full flowering, and Ph3—end of flowering. Different letters indicate significant differences between mean values at *p* ≤ 0.0001.

**Table 5 plants-11-02906-t005:** Pigment compounds content of Virginia mallow and cup plant leaves in different phenophases.

Phenophases	Chl a (mg/g)	Chl b (mg/g)	TCh (mg/g)	TCa (mg/g)
**Virginia Mallow**				
Ph_1_	0.96 ^a^	0.47 ± 0.02 ^b^	1.43 ± 0.02 ^b^	0.37 ^a^
Ph_2_	0.99 ± 0.01 ^a^	0.65 ± 0.03 ^a^	1.64 ± 0.05 ^a^	0.26 ^b^
Ph_3_	0.75 ± 0.03 ^b^	0.56 ± 0.05 ^ab^	1.31 ± 0.07 ^b^	1.31 ± 0.02 ^b^
LSD	0.056	0.110	0.155	0.038
ANOVA	*p* ≤ 0.0001	*p* ≤ 0.0030	*p* ≤ 0.0005	*p* ≤ 0.0001
**Cup plant**				
Ph_1_	0.59 ^a^	0.48 ^a^	1.07 ^a^	0.16 ^a^
Ph_2_	0.58 ± 0.02 ^a^	0.53 ± 0.05 ^a^	1.11 ± 0.07 ^a^	0.15 ^a^
Ph_3_	0.27 ± 0.02 ^b^	0.21 ± 0.02 ^b^	0.47 ± 0.02 ^b^	0.09 ^b^
LSD	0.048	0.086	0.134	0.010
ANOVA	*p* ≤ 0.0001	*p* ≤ 0.0001	*p* ≤ 0.0001	*p* ≤ 0.0001

Ph1—pre-flowering, Ph2—full flowering, and Ph3—end of flowering; Chl_a—chlorophyll a; Chl_b—chlorophyll b; TCh—total chlorophylls; TCa—total carotenoids. Results are expressed as mean ± standard deviation. Different letters indicate significant differences between mean values.

**Table 6 plants-11-02906-t006:** Plant sampling dates for analysis of bioactive compounds.

Phenophase ID	Virginia Mallow	Cup Plant
Ph1—pre-flowering	2 June	17 June
Ph2—full flowering	20 June	27 June
Ph3—end of flowering	13 July	5 September

**Table 7 plants-11-02906-t007:** Equations of calibration curves for individual polyphenolic standards.

Standard	Calibration Curve Equation	R^2^ Value
Caffeic acid	y = 0.2137x − 0.0318	0.9997
Coumaric acid	y = 0.2244x + 2.3792	0.9947
Ellagic acid	y = 0.0338x + 7.0339	0.9309
Ferulic acid	y = 0.3104x + 1.0316	0.9946
Naringin	y = 0.1633x − 0.0833	0.9998

**Table 8 plants-11-02906-t008:** Equations of calibration curves for Trolox standard used for FRAP and ABTS assay.

Antioxidant Assay	Calibration Curve Equation	R^2^ Value
ABTS	y = −0.0002x + 0.614	0.8747
FRAP	y = 0.0007x + 0.0116	0.9958

## Data Availability

Data generated during the study can be obtained by the authors of this study.

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
