# Peer review of "Energy vs. Nutritional Potential of Virginia Mallow (Sida hermaphrodita L.) and Cup Plant (Silphium perfoliatum L.)"

_plants, 2022, doi:10.3390/plants11212906_

Round 1

Reviewer 1 Report

Review – manuscript no. plants-1991348

Review of the manuscript which has been submitted to Plants

Manuscript no. plants-1991348

Title:  Energy vs. nutritional potential of Virgina mallow (Sida hermaphrodita L.) and Cup plant (Silphium perfoliatum L.)

Related to the world energy crisis, the actual context of the study topic entitled “Energy vs. nutritional potential of Virgina mallow (Sida hermaphrodita L.) and Cup plant (Silphium perfoliatum L.)” by Šurić J and co-workers, is well chosen and the article it is very well written. Below I have some suggestions to improve the quality of the work in order to recommend acceptance for publication.

Page 4, row 138: please increase all figures' resolution.

Page 5, row 172: please reformulate the sentence for a better understanding “Low sulfur content was also observed in this study, which is another indication of 172 the potential of the observed crops in direct combustion.” Somehow the information’s repeated.  

Author Response

Dear respectable Reviewer,
Thank you for your valuable comments and suggestions. In the attached document we convey the changes that are also indicated in the main body of the text

Reviewer 2 Report

see the attachment

Author Response

(The authors gave the same response as above.)

Reviewer 3 Report

The paper entitled "Energy vs nutritional potential of Virgina mallow (Sida hermaphrodita L.) and Cup plant (Silphium perfoliatum L.)" describes the energetic, nutritional and functional potential of bioactive compounds in these plants.

The authors have put a lot of effort into the experimental work.

The paper should be carried out the following revisions:

  1. All figures could be improved. The quality of figures is deficient: Figures 1, 2 and 3 – the error bars are not visible; Figure 3 - the units (μmol TE/L) of antioxidant capacity should be on one line;
  2. Section 3.5. High-Performance Liquid Chromatography (HPLC). It would be interesting to see HPLC chromatograms of phenolic compounds profile in all three phenophases. I would recommend adding numbers to the peaks in the chromatogram to more accurately represent their belonging to a particular compound (Figures 5 and 6). The peak number and the corresponding compound can be indicated below the image.
  3. Conclusions. The title contains the word "versus" (vs), which is used when making a comparison between two things, ideas, or opinions. Therefore, in the conclusions and evaluation, the energy potential should be more clearly compared to the nutritional potential; 

Author Response

Dear respectable Reviewer,
Thank you for your valuable comments and suggestions. In the attached document we convey the changes that are also indicated in the main body of the text.
